# A Hydrogel-Based Microfluidic Nerve Cuff for Neuromodulation of Peripheral Nerves

**DOI:** 10.3390/mi12121522

**Published:** 2021-12-08

**Authors:** Raviraj Thakur, Felix P. Aplin, Gene Y. Fridman

**Affiliations:** 1Department of Otolaryngology, Head and Neck Surgery, Johns Hopkins University, Baltimore, MD 21205, USA; ravirajthakur88@gmail.com (R.T.); f.aplin@unsw.edu.au (F.P.A.); 2Department of Biomedical Engineering, Johns Hopkins University, Baltimore, MD 21205, USA; 3Department of Electrical and Computer Engineering, Johns Hopkins University, Baltimore, MD 21205, USA

**Keywords:** neural interface, nerve cuff electrode, peripheral nerve stimulation, direct current nerve block, neural electrode, bioelectronics

## Abstract

Implantable neuromodulation devices typically have metal in contact with soft, ion-conducting nerves. These neural interfaces excite neurons using short-duration electrical pulses. While this approach has been extremely successful for multiple clinical applications, it is limited in delivering long-duration pulses or direct current (DC), even for acute term studies. When the charge injection capacity of electrodes is exceeded, irreversible electrochemical processes occur, and toxic byproducts are discharged directly onto the nerve, causing biological damage. Hydrogel coatings on electrodes improve the overall charge injection limit and provide a mechanically pliable interface. To further extend this idea, we developed a silicone-based nerve cuff lead with a hydrogel microfluidic conduit. It serves as a thin, soft and flexible interconnection and provides a greater spatial separation between metal electrodes and the target nerve. In an in vivo rat model, we used this cuff to stimulate and record from sciatic nerves, with performance comparable to that of metal electrodes. Further, we delivered DC through the lead in an acute manner to induce nerve block that is reversible. In contrast to most metallic cuff electrodes, which need microfabrication equipment, we built this cuff using a consumer-grade digital cutter and a simplified molding process. Overall, the device will be beneficial to neuromodulation researchers as a general-purpose nerve cuff electrode for peripheral neuromodulation experiments.

## 1. Introduction

Implantable neuromodulation devices such as cochlear implants and pacemakers are an important class of medical devices routinely used to stimulate the human nervous system [1]. The neural interface is a critical component of such implants since it is the bridge between ionically conducting nerves and electronically conducting devices. A well-known example of such an interface is a junction formed between metal electrodes such as platinum and a target nerve through a cylindrical cuff type lead, often termed as a “nerve cuff electrode”. Advances in material science and incorporation of *soft electronics* technology continue to improve mechanical properties of neural interfaces and connector leads by making these junctions tissue-like [2]. Several alternate electrode materials such as conducting polymers [3] and nanocomposite coatings [4] have been tested, and a variety of substrate materials and fabrication techniques have been developed for making soft, flexible nerve leads [5]. However, improvements in their functional capacity have largely been overlooked. Conventional neural implants operate predominantly in *excitation mode* where target neurons are stimulated by the delivery of short, charge-balanced, biphasic electrical pulses. While this approach is effective in treating multiple disorders [1], incorporating modalities such as *neural inhibition* can significantly extend clinical applications of these devices [6,7]. Recent studies have shown stimulus waveforms that are not charge-balanced, such as direct current (DC) [8,9] or long-duration electrical pulses, can suppress neural activity in a reversible manner [10,11]. This is important for applications where a smooth gradient of inhibition carries meaningful neural information, such as for vestibular prostheses [12]. Similarly, proof-of-concept studies demonstrate the potential of such waveforms to reversibly block nociceptive fibers within peripheral nerves while potentially maintaining somatosensory and motor conduction, which would allow for targeted pain therapy and management [8,13]. However, there are several technological hurdles for successful clinical translation of these proof-of-concept studies.

One significant challenge is that commercially available neural interfaces do not deliver non-charge-balanced waveforms such as DC in a biologically safe manner. The limitation primarily comes from the neural implant design where a neural interface is typically established by metal electrodes in direct physical contact with target nerve [14,15,16,17,18,19,20]. A typical example is a nerve cuff electrode which is used to stimulate/record from peripheral sciatic nerves [17,20]. It consists of a polymer cylindrical housing that wraps around the nerve and has several openings where metal electrodes establish tight physical nerve contact. As the charge injection capacity of these electrode materials is small (~300 uC/cm^2^ [21]), only short duration pulses can be applied without generation of toxic electrochemical reactions. If the stimulation current waveform is not charge-balanced for these electrode–tissue interfaces, electrochemical reactions such as electrolysis and corrosion start to occur at the electrode surface [21]. Due to its close proximity to the tissue, the toxic byproducts can diffuse rapidly to the nerve causing significant damage or neural death. Additionally, mechanical mismatch between a stiff metal electrode and a soft, viscoelastic tissue has always been a point of concern in terms of long-term physiological safety [2,22]. There remains a need to design a neural interface where biphasic charge-balanced pulses as well as other stimulation modalities such as direct current or monophasic pulses can be safely deployed while minimizing both electrochemical and mechanical damage to the tissue.

A conceptual solution to this problem is to move the metal electrode away from the target nerve while maintaining electrical contact via a fluidic conduit filled with electrolytes. An everyday example of this strategy is electrocardiogram (ECG) recordings, where an ionically conductive hydrogel layer is applied on the skin to keep it hydrated throughout the recording [5]. However, incorporation of such hydrogel conduits for implantable cuff leads is challenging, and prototypes that allow spatial separation while fully satisfying all the other design criterions of clinical implantation have not yet been reported. Some validation studies, for example, Kilgore et al., developed a separated interface nerve electrode (SINE) which consisted of a platinum electrode situated at one end of a saline filled syringe column, while the other end was interfaced with sciatic nerve using a tubing-based cuff [23,24]. This separation is also vital for the function of devices such as the “safe DC stimulator” (SDCS) which rectifies current ionically while maintaining charge balance at the metal interface [25,26,27]. For example, Aplin et al. used a micropipette tube lead design filled with agar gel electrolyte and implanted it in the vestibular canal for DC-based modulation [11]. A similar design has been used for DC block where the conductive agar gel served as an interface with a rat sciatic nerve [13]. Although these in vivo results have validated the utility of non-charge-balanced waveforms such as ionic direct current to induce nerve block, the experimental setup did not emphasize design or fabrication of a lead suitable for implantation and modulation. Moreover, they did not demonstrate whether the same interface can be used more broadly to stimulate using traditional biphasic pulses or record neural activity. While there are a few examples of nerve cuff electrodes equipped with microfluidic channels [28,29,30], their application was solely to deliver chemical drugs either through diffusion or electrophoresis to the nerve. A flexible fluidic lead that can act as an interface between DC delivery devices and nerves in a chronic in vivo setting is a vital step for the translation of these device towards clinical application.

To address this technological gap, we propose a ”microfluidic nerve cuff”, a nerve cuff lead that provides a soft, ionic neural interface using a conducting electrolytic hydrogel. Even though the metal electrodes are placed far from the actual nerve, our hypothesis is that we can stimulate as well as record from rat sciatic nerves with a performance that is comparable to a traditional metal–tissue interface. To reliably meet the general requirements for surgical implantation, we manufactured the cuff to be thin and self-folding. We also present a simple and novel fabrication approach that requires paper and an inexpensive digital cutter. To test the hypothesis that we can pass non-charge-balanced waveforms, we applied a reversible nerve block by delivering cathodic direct current to the sciatic nerve. The development of this proof-of-concept cuff is the first step towards the chronic implantation and interface of devices such as the SDCS so that long-term efficacy and safety of these technologies can be addressed and the concepts moved closer towards clinical translation.

## 2. Results and Discussion

### 2.1. Microfluidic Nerve Cuff Fabrication Using Wet Paper Molds

The microfluidic nerve cuff was manufactured using a soft, elastic silicone material and served as an interconnecting piece between a stainless-steel wire electrode and rat sciatic nerve. Overall, it comprises three functional subunits as seen in Figure 1a: (1) a self-folding cylindrical strip at the tail end of the cuff that wraps around the nerve tightly to establish a robust electrical connection, (2) a set of microfluidic channels carrying electrolytic agar gel that provides an ion-conductive path leading to the nerve and (3) a stainless-steel metallic wire electrode placed in the gel at the front end of the cuff that provides an interface for stimulation as well as recording. As an illustration, Figure 1b,c shows the lead mounted ex vivo on a ~1 mm severed rat sciatic nerve where the channels are filled with a food color mixed with agar gel and a stainless-steel wire electrode is placed approximately 1 cm away from the nerve. The nerve contact area is 110 µm × 700 µm per channel. A cross sectional schematic can be seen in Figure 1b.

The goal of the fabrication process was to create a nerve cuff that has a cylindrical section at the tail end while keeping the overall thickness as minimal as possible so as to abide by general cuff design principles [15,31,32]. Several fabrication methods are documented in the literature for manufacturing polymer nerve cuffs; however, the existing protocols mainly focus on embedding metal electrodes in the cuff housings using microfabrication approaches such as e-beam metal deposition [33,34,35,36]. Since our goal requires microfluidic channels in the order of 100 µm in direct contact with the nerve (due to space and current spread constraints), direct adoption of these methods was not feasible. Our central challenge was to cast microfluidic channels on a thin silicone membrane in a manner that allows easy downstream membrane handling to create the local cylindrical shape. To future-proof the fabrication process for bi- or tri-polar stimulation, and to show that the design was structurally robust even with multiple channels, we designed a tri-channel cuff. Functional tests in the present study only used one channel.

We developed a novel fabrication approach that uses soft, *wet paper* patterns as master molds to cast microfluidic channels in the silicone. The basic principle of this liquid-assisted micromolding process is shown in Figure 2a. Cellulose paper on a hydrophobic acetate transparency film was first cut using a digital vinyl cutter (Figure 2b, steps 1 and 2). Due to its hydrophilic nature, the paper is wetted easily and the added water is absorbed within the bounds of the paper pattern (Figure 2b, step 3). We found that this wet paper matrix serves as a master mold for replica molding of silicone, very similar in function to solid polymer molds that are fabricated using 3D printing or SU8-based lithography [37]. After spin coating silicone premix on the substrate, surface tension forces do not allow infiltration of the silicone into the water filled pores of the paper, resulting in casting reverse features in the cured silicone membrane as shown in Figure 2b (step 4).

The key steps of the paper-based micromolding process for nerve cuff can be seen in Figure 2b. A unique enabling feature of this method is that it allows stretching of the cured membrane with fluidic channels on the mold itself without any rupture. This is because the water is retained in the porous matrix even after curing, causing an easy peel-off. Additionally, the wet paper mold, being mechanically soft, does not impart additional stresses on the stretched membrane, neither on the corners nor on the edges. Figure 2b (steps 4 and 5) shows a prestressed membrane (2× length pre-strain) on the wet paper mold once it is cured. After stretching, subsequent bonding of the fluidic membrane layer to an uncured thin rectangular silicone strip just over the part of the lead that will encircle the nerve yields a cylindrical, self-folding geometry (Figure 2b, step 6) with the width of the strip determining the cylinder radius. Our current nerve cuff design has microfluidic channels that are ~110 µm deep, 700 µm wide and 1.5 cm long with the cylinder radius of ~1 mm as shown. Once the lead is flipped over, the channels are bonded over with unstretched silicone membrane to create the enclosed channels and filled with agar gel (described in detail in the methods), while leaving the electrode contact area in the middle of the folded region unenclosed in Figure 2b (step 7). The self-folding flap of the device that wraps around the nerve is ~50 ± 25 µm, whereas the overall thickness of the flat microfluidic section is ~200 ± 25 µm. The overall thickness of the device and soft nature of the silicone material makes the device flexible as shown in Figure 1d–g and easy to surgically implant around the sciatic nerve in our in vivo animal model as shown in Figure 2b (steps 8 and 9).

The parameters that govern the wet-paper-based molding process are the porosity of the paper, water content and curing temperature. An additional benefit of this manufacturing process is its feasibility in a low-resource setting without the need for expensive microfabrication equipment or injection molding or hot embossing machines. Impedance of the cuff can be a limiting factor when stimulating and recording from distance. Additional characterization of this technique will be the subject of our future work that will quantify channel roughness, resolution and other relevant factors.

### 2.2. Neural Stimulation, Recording and Blocking Using Microfluidic Nerve Cuff

The microfluidic nerve cuff was tested for its neuromodulation efficacy on rat sciatic nerves using the in vivo model as described in the experimental section. We analyzed its performance in three configurations: (1) as a stimulation interface to evoke neural activity, (2) as a recording interface to record signals from the nerve and (3) as an interface to block the neural activity by delivering ionic direct current. In all cases, the distant return electrode was a stainless-steel needle tip inserted into the muscle. As a positive experimental control, we used the stainless-steel needle electrode in direct nerve contact at both the stimulation and the recording sites. We define a neural interface as the electrical interface that exists right at the nerve contact. For example, our microfluidic cuff has an electrolytic gel in direct contact with the nerve, while the primary stainless-steel electrode sits at the other end of that microfluidic channel, away from the nerve. The primary charge carrier is the ionic conducting salts in the electrolytic hydrogel. In contrast, the stainless-steel electrode in direct contact with a nerve is the electronic neural interface in the control condition. The central hypothesis of our experimental design is that an ionic neural interface created by the microfluidic cuff can stimulate as well as record neurophysiological responses and can function similar to an electronic interface provided by a metal–tissue interface. The area of nerve contact was much larger for the microfluidic cuff (110 × 700 µm) than that of a needle electrode tip (~50 µm diameter). In addition, the microfluidic cuff makes circumferential contact with the nerve as compared to a needle tip that pieces into nerve epineurium. As such, we cannot make an objective comparison between the two experimental conditions. Instead, we aimed to demonstrate that we can record and elicit neural responses using the microfluidic nerve cuff at noise/current levels comparable to a standard metal–tissue interface.

Figure 3 shows examples of the electrical waveforms recorded in response to biphasic pulse stimulation for the stainless steel (a) and microfluidic (b) interfaces. Figure 3c shows the example waveforms of the microfluidic nerve cuff used at the recording site on the nerve. Figure 3d shows the nerve cuff deployed as a device to deliver cathodic ionic direct current (iDC) to block the evoked neural activity.

Two electrical peaks were consistently found in all neural recordings for every animal: one emerging from the evoked compound action potentials (ECAP) occurring ~500 µs from stimulation onset and the other being electromyographic (EMG) peaks occurring ~2 ms from the onset. Both the ECAP and EMG peak amplitudes progressively increased with increasing stimulation current. A quantitative comparison between the two stimulation methods is made in Figure 4, which plots the average ECAP and EMG voltage peaks recorded from four animals for different values of the stimulation current. From the plots, we observed that EMG responses were saturated at approximately 400 µA for the stainless-steel stimulation electrode as seen in Figure 4a,b, whereas the microfluidic cuff saturation occurred at approximately 150 µA as seen in Figure 4d. This observation suggests that the current design produces a robust, insulated electrical contact that is consistently established by the self-folding flap of the cuff as opposed to a leak-prone interface provided by the needle electrodes. ECAP signals however were not saturated as seen in Figure 4b,e when stimulated with the stainless-steel needle electrode. We believe this is mainly because of our experimental protocol. Since we increased the stimulation current in a stepwise manner, most muscle fibers recruited undergo fatigue first, and the EMG signal thus tapered off before ECAP. We believe the ECAP saturation could have been visible if we had used higher (>400 µA) stimulation currents.

From Figure 4c,f, it can be seen that all responses are completely suppressed by the delivery of an anesthetic drug, lidocaine, which confirms that the recorded waveforms are not a measurement artifact and indeed stem from the evoked neural and muscle activation. Since our experimental protocol was performed on terminal, decapitated rats, both ECAP/EMG signals were subject to natural decay during the time course of every experiment. Thus, addition of lidocaine was performed as the last step of the protocol to serve as the experimental control. The ECAPs recorded from the microfluidic cuff were found to be an order or magnitude higher than the voltages recorded from the needle electrode as seen in Figure 4h. As EMG signals recorded were analogous to those of the control, we believe this extra sensitivity towards recording ECAPs is additional evidence of the robust electrical contact at the nerve. We also noticed that the recorded signals were noisier compared to the control as seen in Figure 3a,c. This is likely due to a higher channel impedance of the long microfluidic hydrogel channel where ion mobility is substantially lower compared to electrons. This is one of the limitations of this method because the voltage requirement is higher for this nerve cuff. For example, our microfluidic channel resistance was ~100 kΩ/cm (measured using an impedance analyzer). To deliver a current of 100 µA, we needed a supply voltage of 10 V.

As a proof-of-concept application of the microfluidic nerve cuff, we demonstrate an acute nerve block induced on rat sciatic nerve. For this protocol, monopolar stainless-steel needle electrodes were used to send stimulation pulses and to record from the sciatic nerve. From the sample waveforms, we see that both the ECAP and EMG peaks are progressively suppressed with the increasing amplitude of iDC delivered through the microfluidic cuff. Figure 4j,k shows the plots for the recorded ECAP and EMG peak-to-peak voltages as a function of the DC amplitude. We found that the neural signals were partially blocked for the iDC values under 75 µA, whereas for the larger values a complete block was established. This threshold for block is substantially lower compared to the values reported in recent literature [13,24] emphasizing the potential utility of this device for this application. Furthermore, we observed that both ECAP and EMG signals were recovered immediately after the block was withdrawn as seen in Figure 4l. Complimentary to the electrical recordings, we observed that the electrically evoked muscle movement completely stopped when iDC was applied and returned when it was removed indicating that nerve cuff was reversible and safe for the current procedures. In contrast, DC passed through a stainless-steel electrode in direct contact with the nerves resulted in an irreversible loss of neural activity (data not shown, n = 2), likely due to rapid pH changes and formation of bubbles due to electrolysis at the metal–tissue interface. These results support our hypothesis that having a longer ionic channel delays the diffusion of toxic byproducts, and DC can be delivered in a safe manner for a short time. Throughout our experiment, the cuff lead remained securely fastened to the nerve even in the presence of twitching, and no motion artifacts were observed in the neural recordings.

The long-term mechanical stability, safety and biocompatibility of this cuff will need to be determined and will be the subject of our next experimental studies. Chronic implantation will require further improvements in the design. We foresee that the cuff design would need to change to allow it to move freely with the nerve laterally within the muscle. For this reason, anchoring the cuff to the muscle may actually prevent the natural lateral motion within the muscle. Instead, we envision that the microfluidic channels would be oriented not perpendicularly to the nerve as they are in this prototype, but would continue along the nerve, exiting the muscle parallel to the nerve. This would allow for free movement of the nerve laterally and allow the cuff to move with it rather than impeding its motion.

While our electrode was designed for multiple contacts in principle, we limited our experimental investigation exclusively to monopolar function to simplify the experimental design. Because the contacts are located very near each other at the nerve interface, they may connect and shunt at the interface. To improve this connection, we propose that a soft insulating gel be positioned between the individual contact points on the nerve. These could be made from multiple possible hydrophobic materials, including PDMS or alternative oil-based polymers.

## 3. Experimental Methods

### 3.1. Microfluidic Nerve Cuff Fabrication Using Wet Paper Molds

The fabrication of the microfluidic nerve cuff was performed by combination of two methods: the creation of microfluidic channels in the flat part of the cuff followed by a protocol that imparts the self-folding, cylindrical shape. We developed a novel replica molding method that uses wet paper molds to cast microfluidic channels on a thin, spin coated silicone membrane and subsequently stretch and bond it to an uncured silicone piece to create a local self-folding cylindrical geometry. The overall fabrication protocol is as follows: First a sticker paper (Avery Products Corporation, Brea, CA, USA) was placed on a transparent acetate transparency film (Staples, Framingham, MA, USA) making sure there are no bubbles. A CAD program was used to create a rectangular microfluidic channel of desired dimensions, and the paper was fed into a digital vinyl cutter (Cricut Explore Air, Cricut, South Jordan, UT, USA). After the cutting operation is performed, the background paper was peeled off carefully, leaving behind the paper molds. A 3 channel CAD design (700 µm width, 15 mm length) and the resulting paper mold from the process are seen in Figure. Next, 50 µL of water was added over the paper molds, and after 1 min excess water was blotted off using a blotting paper, making sure the paper is fully saturated with water (Figure 2b, step 3). We used a fast cure silicone (Ecoflex 00-35 Fast Platinum Cure Silicone Rubber Compound Kit, (Smooth-on Macungie, PA, USA) as the base material for making the nerve cuff. A 1:1 mixture of the two-part prepolymers was mixed gently, poured on top of the wet paper molds and spin coated at 500 rpm for 30 s. After incubating for 10 min at room temperature, the membrane was cured, and microfluidic channels were cast on this thin membrane. Then, the cured membrane was peeled off partly from the transparency film and stretched to approximately twice its length over the same mold (Figure 2b (step 4)). To create an uncured silicone strip, we spin coated a new polymer mixture at 500 rpm on a glass slide covered with a thin parafilm. The coating with parafilm facilitates an easy release substrate for this strip. A rectangular strip of ~5 mm width and 3 cm length was then cut and pasted immediately on the stretched membrane while it was still in the uncured state. This step is time sensitive because of the fast curing time of the silicone kit used. After incubation for another 10 min, the parafilm layer was peeled off leaving behind the cured strip bonded to the back of the nerve-contact portion of the stretched membrane. In the next step, the membrane with fluidic channels was slowly relaxed and automatically gained a curled shape where the strip had been bonded as shown in Figure (step 5). The pre-strained amount determines the radius of curvature of the curled part of the cuff, whereas the membrane strip width determines the length of the curled flap. For example, higher strain resulted in decreasing the nerve cuff diameter. The current manufacturing parameters were found after performing several iterations to yield the cuff that would fit ~1 mm rat sciatic nerves. The self-folding is a result of the elastic spring-like nature of the thin silicone membrane. In the final fabrication step, another featureless membrane was spin coated on a paraffin coated glass slide and bonded to this assembly to enclose all the microfluidic channels. This results in microfluidic channels that are closed at the front end while leaving the opening into the cylindrical section of the device. Before an in vivo experiment, the excess silicone material was cut, and the whole assembly was lifted off from the transparency base and checked for self-folding function (Figure 2b, step 6). To prime the device, the conductive 5% agar gel mixture (detailed below) was liquified using a microwave oven and quickly loaded in a syringe with a 36-gauge needle. The needle was inserted in the free end of the microfluidic channel, and liquified gel is slowly flowed out towards the cuff section. After incubating for 5 min, the gel cooled and solidified inside the microfluidic channels. Finally, a stainless-steel wire electrode was cut and inserted at the free end of the device (Figure 2b, step 7).

### 3.2. Animals and Surgical Preparation

Experiments were performed on post-mortem adult Sprague Dawley rats (n = 4) at the terminal endpoint of unrelated studies. All procedures were approved by the Johns Hopkins Animal Care and Use Committee (protocol RA19M306). Animals were placed under anesthesia (Isoflurane, Baxter, Deerfield, IL, USA) and decapitated using a guillotine. The left or right sciatic nerve was then exposed at random and submerged in Kreb’s solution (in mM: 117.9 NaCl, 4.7 KCl, 25 NaHCO_3_, 1.3 NaH_2_PO_4_, 1.2 MgSO_4_, 2.5 CaCl_2_ and 11.1 D-glucose, pH balance: 7.4). Metal electrodes (27-gauge stainless-steel needles) were gently inserted into the exposed sciatic nerve at the most distal and proximal exposed points ~3 cm apart. Two identical electrodes were also inserted into muscle (gluteus maximus for stimulation return, biceps femoris for recording) to act as returns. The microfluidic cuff, filled with Kreb’s solution suspended in 5% agar, was then wrapped around the sciatic nerve between the two metal electrodes (see Figure 2b, step 7). Directly before stimulation the solution around the nerve was drained and replaced with mineral oil to improve electrical insulation within the setup.

### 3.3. Nerve Stimulation and Recording

Sciatic nerve stimulation was achieved using 100 µs, 50–500 µA anodic-first charge-balanced biphasic pulses delivered via an AM2100 isolated pulse stimulator (A-M Systems, Sequim, WA, USA) to the proximal electrode or microfluidic cuff. DC block was achieved through the microfluidic cuff using 12 s, −10 to −125 µA current steps via a Keithley 6221 precision current source (Tektronix, Timonium, MD, USA). To reduce the sharp rise time of the DC step, a 0.01 µF capacitor was placed in parallel with the circuit that introduced a time constant of 10 ± 2 ms. Differential recordings from the distal electrode or microfluidic cuff to the reference electrode were band-pass filtered through a pre-amplifier (P55 A.C. Pre-Amplifier, Grass Instrument CO., West Warwick, RI, USA) between 100 Hz and 10 kHz and amplified at 100x gain before acquisition with a Micro1401 DAQ (Cambridge Electronic Design, Cambridge, UK). All recordings were averaged across 10 repetitions spaced 1 s apart. “Pre-block”, “at block” and “post block” were defined as the 10 stimulation repetitions directly before, during and after the DC current step, respectively. As tissue responses decreased over time, stimulation amplitude and protocol order were varied pseudo-randomly to reduce timing effects. As a positive control, 100 µL lidocaine (10 mg/mL, Hospira INC., St Laurent, QC, Canada) was injected around the nerve at the recording site, with excess fluid removed to prevent current shunting through the lidocaine vehicle. Sciatic nerve stimulation was then recorded immediately following lidocaine injection and at 1–2 min post-injection. The evoked action potentials were found to lie within 0.2 to 5 ms from the time of stimulation. The main reason for the small timing is that the distance between the microfluidic cuff and the recording electrode was rather short, approximately 1.5–2 cm. The conduction velocities for a rat sciatic nerve range between 15 and 45 m/s [38]. The approximate duration to travel 1.5 cm is therefore between 0.3 and 1 ms. Accounting for biological variability, our measurement of the ECAP/EMG falls approximately within this range. The amplitude of the CAP and EMG response was measured from peak to peak of the waveform.

## 4. Conclusions

In this work, we present a proof-of-concept microfluidic nerve cuff which provides a soft, hydrogel-based neural interface. Through experiments conducted in the in vivo rat sciatic model, we validated its functionality as an interconnecting neuromodulation device between a target nerve and a distant metal electrode to stimulate and to record neurophysiological signals. We also showcase reversible nerve block by delivering iDC via the cuff for a short amount of time. We developed a simple fabrication method through a novel concept of wet-paper-based micromolding that would enable researchers in low resource settings to manufacture this device easily, without the need for expensive microfabrication facilities. Our microfluidic cuff provides a missing component that, in conjunction with previously described methods for delivery of DC to neural tissue such as SDCS or SINE, facilitates delivery of DC to tissue while potentially maintaining biological safety. One of the limitations of this approach is that the electrical impedance of the device is higher than metal electrodes alone owing to the low mobility of ions in the microfluidic channel and could provide challenges in power management in certain applications. However, we believe the flexibility, ease of manufacture and robustness of this design will be advantageous over the conventional cuff electrodes in neuromodulation applications that require long durations of current delivery. The immediate next step for this technology is the testing of chronic safety and efficacy of these leads coupled with SDCS systems for the treatment of neuropathic pain.

## Figures and Tables

**Figure 1 micromachines-12-01522-f001:**
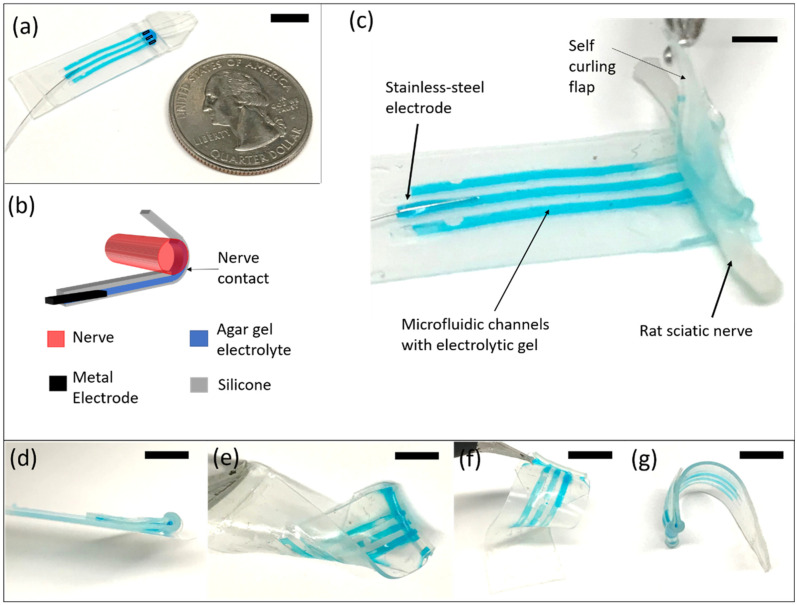
(**a**) The microfluidic nerve cuff for stimulating and recording of sciatic nerves from a distance. Channels open to the nerve are artificially indicated in black. (**b**) Schematic cross-sectional view of the cuff. The long microfluidic conduit containing agar gel electrolyte is open at the end, establishing a firm but soft contact with the nerve. (**c**) A zoomed in image of the nerve cuff mounted on a severed rat sciatic nerve of ~1 mm diameter. The self-curling silicone flap ensures a proper electrical contact while the electrolytic gel provides a soft neural interface. The metal electrode can be placed at a distance from the point of nerve contact. The image shows three microfluidic channels that are open at the end. (**d**–**g**) Images taken from several angles showing the thin and flexible nature of the silicone nerve cuff design. All scale bars represent 5 mm.

**Figure 2 micromachines-12-01522-f002:**
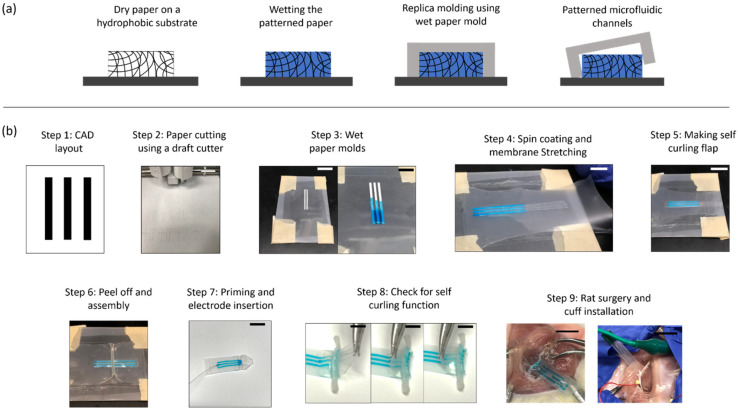
Fabrication of the microfluidic nerve cuff using liquid assisted micromolding of silicone. (**a**) Schematic showing conceptually how wet paper can be used as a master mold for making microfluidic channels in silicone. Since water and silicone are immiscible, wetted hydrophilic paper patterns ensure an easy peel off from the cured polymer. (**b**) Workflow detailing several important steps in the microfluidic nerve cuff fabrication, from computer-aided design (CAD) layout of the electrolyte conduits to the in vivo installation of the nerve cuff on a rat sciatic nerve. Most notably, the wet paper mold permits stretching of a thin silicone membrane on the mold itself as shown in step 4, a function that is otherwise difficult to perform on solid molds. The scale bar represents ~5 mm.

**Figure 3 micromachines-12-01522-f003:**
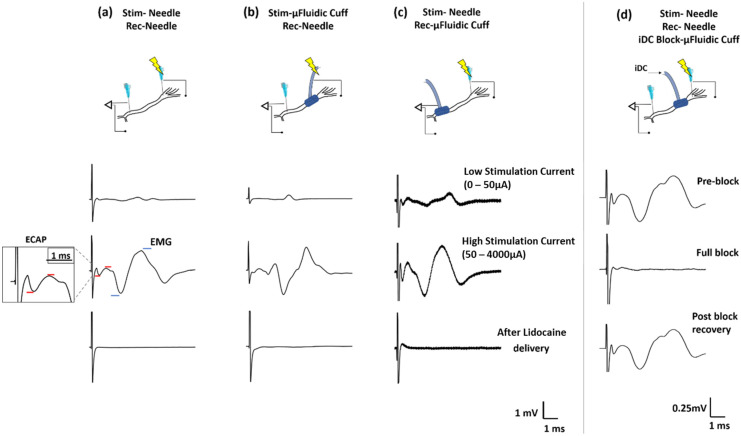
Sample in vivo neurophysiological recordings from the rat sciatic nerve model for different test conditions: (**a**) stainless-steel needle serving both as stimulation and recording electrode, (**b**) stimulation through the microfluidic nerve cuff and recording through the needle electrode, (**c**) stimulation through the needle electrode while recording through the nerve cuff. Three sample recordings for each case show the electrical signal recorded below 50% saturation and at saturation value of the stimulation current as well as the effect of lidocaine delivery on the sciatic nerve. Both evoked compound action potential (ECAP) as well as electromyographic (EMG) peaks can be clearly identified from the recordings. (**d**) Sample recordings show that the neural activity is completely arrested by the application of direct current delivered through the microfluidic nerve cuff.

**Figure 4 micromachines-12-01522-f004:**
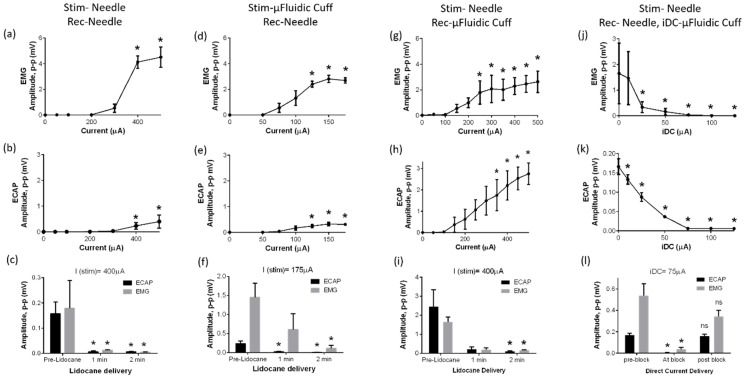
Quantification of peak-to-peak EMG and ECAP responses for different experimental conditions. (**a**–**c**) Plots showing responses when stainless-steel electrodes were used for stimulation and recording. (**d**–**f**) Plots showing stimulation through the microfluidic nerve cuff and recording from the needle electrode. (**g**–**i**) Plots showing stimulation through a needle electrode while recording through the microfluidic cuff. We observed that both the EMG and ECAP responses increased as we progressively increased the stimulation current amplitude. After the delivery of lidocaine, all responses were suppressed. (**j**–**l**) Cathodic direct current (DC) delivered to the nerve through the microfluidic nerve cuff while stimulating and recording through the needle electrodes. Both the EMG and ECAP decreased continually with the magnitude of the direct current delivered and were recovered after removal of the DC block. * *p* < 0.05 was chosen to establish statistical significance for n = 4 using a one way repeated measures ANOVA with Dunnett’s multiple comparison test versus baseline.

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
