# Peer review of "A Hydrogel-Based Microfluidic Nerve Cuff for Neuromodulation of Peripheral Nerves"

_micromachines, 2021, doi:10.3390/mi12121522_

Round 1
Reviewer 1 Report
The author demonstrates a hydrogel-based microfluidic nerve cuff for neuromodulation of peripheral nerves. There are serious issues to be addressed.
First, the author did not conduct essential experiments. For instance, the author should conduct EIS and CV tests to evaluate the electrical performance of the devices. How much the impedance at 1 kHz and how much the impedance when to implant on a nerve? How much CSC and CIC?
Second, what is the thickness of the device? and what is self folding mechanism and how to control depending on the size of nerves?
Third, did the author measure conduction velocities? The amplitudes of CNAPs are around a few hundreds volt and the duration is also quite small (around 1~2 ms). The author need to provide exact values of CNAP aplitude. For instance, Fig 4(f) shows more than 1~3 mV amplitudes.
Author Response
The author demonstrates a hydrogel-based microfluidic nerve cuff for neuromodulation of peripheral nerves. There are serious issues to be addressed.
First, the author did not conduct essential experiments. For instance, the author should conduct EIS and CV tests to evaluate the electrical performance of the devices. How much the impedance at 1 kHz and how much the impedance when to implant on a nerve? How much CSC and CIC?
Author’s response- While we recognize both CV and EIS have been used in characterizing neural interfaces, these methods are used to measure the electrical performance of metal electrodes in contact with the nerve. In our design, metal electrode is separated from the nerve through a microfluidic channel. The experiments conducted in this paper used a stainless-steel electrode, however any other electrode materials can be used such as platinum or carbon black with different surface areas. This manuscript does not specify the method of interfacing to the cuff other than to conduct the ionic current flow. The goal of our experiments is to evaluate the functional performance of hydrogel-nerve interface and not the metal electrode interface to the hydrogel. For this reason we believe CV or EIS study will not yield useful information. The overall impedance of the electrode and the microfluidic channel in the cuff was measured to be ~100kOhms as described in the line 265 on page 8.
Second, what is the thickness of the device? and what is self-folding mechanism and how to control depending on the size of nerves?
Author’s response-
The total thickness of the device is ~200 +/- 25 µm as indicated in the line 184, Page 5.
The pre-strained amount determines the radius of curvature of the curled part of the cuff whereas the membrane strip width determines the length of the curled flap. For example, higher strain resulted in decreasing the nerve cuff diameter. The current manufacturing parameters were found after performing several iterations to yield the cuff that would fit ~1mm rat sciatic nerves. The self-folding is a result of the elastic spring-like nature of the thin silicone membrane. We have added this explanation to the revised manuscript line 352-361 on Page 10.
Third, did the author measure conduction velocities? The amplitudes of CNAPs are around a few hundreds volt and the duration is also quite small (around 1~2 ms). The author need to provide exact values of CNAP aplitude. For instance, Fig 4(f) shows more than 1~3 mV amplitudes.
Author’s response-
The main reason for the short CNAP timing is that the distance between the microfluidic cuff and the recording electrode was rather short, approximately 1.5- 2 cm. The conduction velocities for a rat sciatic nerve range between 15-45 m/s. The approximate duration to travel 1-1.5cm is then 0.45-2ms. Accounting for biological variability, our measurement of the CNAPs fall approximately within this range. We have added this brief explanation in the manuscript on line 404-412 on page 11.
Fig 4 as a whole shows the exact values of both ECAP and EMG signals that were measured for different experimental conditions. Fig 4(h) and (i) show 1-3mV ECAP amplitude when measured from our microfluidic cuff electrode as opposed to only 0.175mV when measured from a stainless-steel electrode. We believe, this is a rather distinct advantage of our device as it is almost an order of magnitude more sensitive compared to metal electrode for measuring ECAP. This explanation can be found on line 257-260 on page 8.
Reviewer 2 Report
This paper is extremely well written and clear. The protocol is well thought out and complete. Just a few questions.
-Why do the ECAP/EMG signals decay? Is this possibly damage to the nerve?
-I didn't see the description of the 5% agar gel mixture.
Author Response
Reviewer 2
This paper is extremely well written and clear. The protocol is well thought out and
complete. Just a few questions.
Thank you.
-Why do the ECAP/EMG signals decay? Is this possibly damage to the nerve?
Author’s response: Our experiments were performed on decapitated rats thus both ECAP/EMG signals were subject to natural signal decay during the time course of every experiment. We used lidocaine in the last step of the protocol to serve as the experimental control to ensure that the responses of any size were in fact biological in nature rather than artifactual.
-I didn't see the description of the 5% agar gel mixture.
Author’s response: This process has been described in the experimental section, 3.2 Animal and Surgical preparation, line 374-378.
Reviewer 3 Report
The submitted manuscript is detailing the design of a nerve cuff that incorporates a gelatonous platform to improve the neural interface when stimulating major nerves for wanted effects. Conceptually this is a much needed advancement in the field given how vast the use of nerve stimulation is in both preclinical and clinical applications. The research teams delineates their design, methods, and results nicely in this well written manuscript. They also use the appropriate controls when testing their cuff. I look forward to what they are able to produce for more broad use. This truly could revolutionize the field of neuromodulation.
Author Response
Reviewer 3
The submitted manuscript is detailing the design of a nerve cuff that incorporates a gelatonous platform to improve the neural interface when stimulating major nerves for wanted effects. Conceptually this is a much needed advancement in the field given how vast the use of nerve stimulation is in both preclinical and clinical applications. The research teams delineates their design, methods, and results nicely in this well written manuscript. They also use the appropriate controls when testing their cuff. I look forward to what they are able to produce for more broad use. This truly could revolutionize the field of neuromodulation.
Author’s response: Thank you!
Reviewer 4 Report
A Hydrogel-based Microfluidic Nerve Cuff for Neuromodulation of Peripheral Nerves
This is a well written manuscript on a fluid based electrode system.
Major comments:
- I believe that the electrode shape will not work for chronic implantation since the curl is too open and flexible and it will not maintain its position. Please discuss this and possible improvements in the design.
- The agar gel at the end openings of the three micro tubes have a high chance of bleeding into each other, which would short circuit the electrode and prevent bipolar stimulation and/or recording. The authors have used monopolar activation and recording but not bipolar. I think it is important to discuss this in the manuscript.
Minor comments;
Figure 1. Please indicate with a different color where the microfluid channel is open at the nerve end. Figure 1 a seems to show that the 3 channels are in contact with each other at the nerve end.
Figure 3. The spelling of Lidocaine is not Lidocane.
Line 333: Should be Gluteus Maximus
Author Response
This is a well written manuscript on a fluid based electrode system.
Thank you!
Major comments:
1. I believe that the electrode shape will not work for chronic implantation since the curl is too open and flexible and it will not maintain its position. Please discuss this and possible improvements in the design.
2. The agar gel at the end openings of the three micro tubes have a high chance of bleeding into each other, which would short circuit the electrode and prevent bipolar stimulation and/or recording. The authors have used monopolar activation and recording but not bipolar. I think it is important to discuss this in the manuscript.
Thank you for these observations. We agree. We introduced the following text to the end of the Results and Discussion.
Chronic implantation will require further improvements in the design. We foresee that the cuff design would need to change to allow it to move freely with the nerve laterally within the muscle. For this reason, anchoring the cuff to the muscle may actually prevent the natural lateral motion within the muscle. Instead, we envision that the microfluidic channels would be oriented not perpendicularly to the nerve as they are in this prototype, but continue along the nerve, exiting the muscle parallel to the nerve. This would allow for free movement of the nerve laterally, and allow the cuff to move with it rather than impeding its motion.
While our electrode was designed for multiple contacts in principle, we limited our experimental investigation exclusively to monopolar function to simplify the experimental design. Because the contacts are located very near each other at the nerve interface, they may connect and shunt at the interface. To improve this connection, we propose that a soft insulating gel be positioned between the individual contact points on the nerve. These could be made from multiple possible hydrophobic materials, including PDMS or alternative oil-based-polymers.
Minor comments;
Figure 1. Please indicate with a different color where the microfluid channel is open at the nerve end. Figure 1 a seems to show that the 3 channels are in contact with each other at the nerve end.
Figure 3. The spelling of Lidocaine is not Lidocane.
Line 333: Should be Gluteus Maximus
These are fixed. Thank you.
Round 2
Reviewer 1 Report
The author addressed almost all review comments well. But, there are still spelling errors of Lidocane in Fig. 4.